# A Cross-Sectional Comparison of Physical Activity during COVID-19 in a Sample of Rural and Non-Rural Participants in the US

**DOI:** 10.3390/ijerph18094991

**Published:** 2021-05-08

**Authors:** Alan M. Beck, Amanda S. Gilbert, Dixie D. Duncan, Eric M. Wiedenman

**Affiliations:** 1Prevention Research Center, Washington University in St. Louis, St. Louis, MO 63130, USA; a.s.gilbert@wustl.edu (A.S.G.); dixieduncan@wustl.edu (D.D.D.); ericw@wustl.edu (E.M.W.); 2Department of Surgery, Division of Public Health Sciences, Washington University in St. Louis, St. Louis, MO 63130, USA

**Keywords:** physical activity, COVID-19, rural

## Abstract

Physical activity (PA) pre-COVID-19 was lower in rural areas compared to non-rural areas. The purpose of this study was to determine COVID-19’s impact on PA in rural and non-rural residents. A cross-sectional study consisting of a convenience sample of 278 participants (50% rural, 50% non-rural) from 25 states completed an online survey describing their PA behaviors and perceptions during COVID-19. The global physical activity questionnaire was used to determine PA in various domains and summed to determine if the participant met the PA guidelines. Rural participants had a significantly higher body mass index, lower income, and a lower educational attainment. Conversely, non-rural participants reported more barriers to PA. There was no difference in the perception of COVID-19’s impact on PA, specifically; however, rural participants were significantly less likely to meet cardiorespiratory PA recommendations compared to non-rural participants. Conclusions: This study demonstrates the continued disparity in PA between rural and non-rural residents, despite the supposition of COVID-19 being less impactful in rural areas due to sparse populations. Efforts should be pursued to close the PA gap between rural and non-rural residents.

## 1. Introduction

People in rural communities tend to be older, obese, have higher rates of tobacco smoking, and have more pre-existing comorbidities compared to people in urban areas [1,2,3,4,5]. These same characteristics, smoking, obesity, and pre-existing comorbidities, are also associated with poor outcomes in COVID-19 such as hospital admission, mechanical ventilation, and death [6,7,8,9]. Furthermore, 80% of mortalities due to COVID-19 are among adults over the age of 60 [10,11]. More than one in five older Americans live in rural counties and the percentage of individuals over the age of 65 increases with rurality [12]. These characteristics may suggest that rural areas in the US are more at risk for poor outcomes related to COVID-19, both due to common health behaviors and overall age of the rural population [8]. 

To combat COVID-19 in rural areas, the Centers for Disease Control and Prevention (CDC) recommends improvements in overall health [13]. Physical activity (PA) has the ability to prevent and treat chronic diseases and obesity [14,15,16]. Recent studies have suggested that engaging in regular PA may improve clinical conditions most commonly related with severe COVID-19 [17,18,19]. Conversely, physical inactivity increases the risk of poor long-term health outcomes due to COVID-19 infection [15,16,19,20,21]. COVID-19 has had a dramatic negative impact on worldwide PA; however, the data were specific to the largest cities (including US cities) [22,23]. Limited data exist exploring the impact of COVID-19 on PA levels in rural areas, but historically rural areas report lower levels of PA as compared to their non-rural counterparts [24,25]. Without fully understanding how COVID-19 has influenced PA in rural areas, it is difficult to make concerted recommendations.

### Aim of the Current Study

The aim of this study was to determine COVID-19’s impact on meeting PA guidelines in rural communities, non-rural communities, and a comparison between community types. 

## 2. Materials and Methods

### 2.1. Setting

Potential participants were recruited via social media posts (e.g., Facebook) and email list serves (available from 26 August to 17 September 2020). Inclusion criteria for participation was ≥18 years of age, ability to read, and the ability to be physically active (i.e., no limitations barring physical activity). For social media posts, an image describing the study was posted with a link to the survey imbedded in the image. Social media posts were posted to regional and national rural advocacy groups’ social media pages. For email correspondence, a brief description was provided along with a link to the survey. All recruiting materials explicitly asked for individuals living in rural areas to complete the survey. When the potential participant clicked the link, they were routed to a consent form—if they consented to the survey, they would then begin the survey. All data were collected in Qualtrics survey software [26]. The Institutional Review Board at Washington University in St. Louis approved this study with an exempt status. 

### 2.2. Measures

#### 2.2.1. Demographics

The survey asked participants to provide their age, gender, marital status, highest attained education, race, ethnicity, height, weight, income, and zip code. Age, height, and weight were collected as continuous variables with open-ended responses from participants. Categorical variables included gender (male/female), marital status (married, divorced, separated, never married, widowed, living with a partner), highest education obtained (≤8th grade, some high school, completed high school or equivalent, business/technical school, some college, Associates degree, Bachelor’s degree, Master’s degree, beyond Master’s), race and ethnicity (white, black or African American, Hispanic, American Indian or Alaska Native, Native Hawaiian or Pacific Islander, Asian), income (<USD 10,000, USD 10,000–USD 19,000… >USD 90,000), and zip code included open-ended responses. Body mass index (BMI) was calculated with the height and weight variables using the standard BMI formula. BMI was then categorized into underweight (<18.5 kg/m^2^), normal weight (18.5–24.9 kg/m^2^), overweight (25.0–29.9 kg/m^2^), and obese (≥30 kg/m^2^). Age was also categorized into three age groups (≤35, 36–60, and >60 years of age). Using the participants’ zip code, we were able to define rurality by using the associated Rural–Urban Continuum Codes (RUCC) [27], and a code of four or more was considered rural. While participants were explicitly asked to complete the survey only if they lived in a rural area, the RUCC variable was included to provide an objective and scaled measure of rurality. 

#### 2.2.2. Physical Activity

PA was measured using the global physical activity questionnaire (GPAQ) [28]. The GPAQ assesses PA on various constructs (i.e., work, recreational, commute) and intensities (i.e., moderate and vigorous). An example question is “do you do any vigorous-intensity sports, fitness or recreational (leisure) activities that cause large increasing in breathing or heart rate like running or football for at least 10 min?” If the participant answered “yes”, two follow-up questions would come up—“in a typical week, on how many days do you do vigorous intensity sports, fitness or recreational (leisure) activities?” and “how much time do you spend doing vigorous intensity sports, fitness or recreational activities on a typical day?” If the participant answered “no” to the question regarding participation, no further questions were asked about that specific domain, and a score of zero minutes in the domain was assumed. The moderate and vigorous intensity activities were summed individually to determine if participants met cardiovascular PA guidelines (i.e., 150 min of moderate intensity and/or 75 min of vigorous activity) [14]; a dichotomous variable was then computed based on the participant meeting, or not meeting, PA guidelines [29]. The dichotomous meeting guidelines variable was used as the dependent variable. Participants were asked about how the pandemic had influenced their PA by asking, “did your normal physical activity change due to the COVID-19 pandemic?”—if yes, questions about frequency, intensity, and time were further asked and dichotomized into “more” or “less”. It is important to note, the GPAQ in this study had no missing answers due to the requirement to answer the questions due to a forced response; meaning, participants had to answer all of the GPAQ questions prior to moving on due to the skip pattern. While helpful from a data perspective, a forced response led to skewed results whereby people were likely to underestimate their PA. 

#### 2.2.3. Perceptions and Access

Survey items to measure perceptions of how COVID-19 impacted access and PA were developed by the research team. Participants were asked “Which of the following reasons have impacted your access to places for physical activity during the COVID-19 pandemic?” and could check all that applied in the provided list (i.e., gym or fitness center, local trails, parks, team sports). Participants were also asked “Which of the following reasons, if any, have negatively impacted your physical activity during the COVID-19 pandemic?” and could check all applicable options listed (wearing a cloth face covering/mask when outside, time, stress, size or layout of indoor space, size or lack of yard, spaces available or open in my neighborhood, safety of my neighborhood, motivation or interest, concern for exposure to COVID-19, or other). Those who did not answer specific questions were excluded from the analysis. 

### 2.3. Data Analysis

Means, standard deviations, and percentages were calculated for all demographic characteristics. Independent samples *t*-tests and Mann–Whitney U tests for skewed variables were used to determine differences between group characteristics. Chi-squared tests were used to determine differences in meeting PA guidelines both within and between rural and non-rural participant groups. For statistically significant correlations, logistic regressions were conducted to determine the association between demographics (age category and obesity category), perceptions and access with meeting PA guidelines within rural and non-rural participant groups. For example, logistic regressions were run separately by age category and by obesity category for perceptions of COVID-19 impact on PA with meeting PA guidelines for rural participants. Additionally, logistic regressions were run for significant correlations for non-rural participants. Statistical significance was assessed at an alpha of 0.05. All data were analyzed using SPSS® 25.0 for Windows® (IBM Corporation, Armonk, NY, USA). 

## 3. Results

A total of 365 people took part in the survey; however, 87 did not include a zip code and/or PA measures, and so they were removed from the analysis. Of the remaining 278 participants who completed the survey, precisely half were rural (n = 139) and half were non-rural (n = 139). The survey respondents used for analysis represented 25 states. Most of the non-rural and rural respondents were from RUCC three (i.e., counties in metro areas of fewer than 250,000) and six (i.e., 2500–19,000 adjacent to a metro area), respectively. The participants were mostly white, married, female, with a BMI in the category of “overweight”, and made over USD 50,000 per year (Table 1). 

### 3.1. Between-Group Results

People from rural areas were significantly more likely to have a higher BMI (*M =* 30.73, SD = 7.29 (“obese category”)) than non-rural people (*M =* 28.6, SD = 7.19 (“overweight category”), *p* = 0.017, *d =* 0.29). Rural people were significantly less likely to meet PA guidelines (*p =* 0.005), have an income above USD 50,000 (*p =* 0.009), and hold a Bachelor’s degree (*p <* 0.001) (Table 1). While no change was noted in rural residents (*x*^2^(4) = 2.257, *p* = 0.689), non-rural residents did experience a statistically significant correlation in the amount of time being physically active and meeting PA guidelines due to COVID-19 (*x*^2^(4) = 10.072, *p* = 0.039). We also found a statistically significant correlation between not being able to participate in team sports due to concern about exposure to COVID-19 with meeting PA guidelines for non-rural participants (*x*^2^(1) = 5.604, *p* = 0.018) and not for rural participants (*x*^2^(1) = 1.208, *p* = 0.272). Due to the skewed nature of the PA data in our sample, the Mann–Whitney U test was used for the analysis. For the various domains of PA (Table 2), non-rural participants averaged significantly more minutes of vigorous recreational PA (U = 8100.50, N1 = 139, N2 = 139, *p =* 0.006). When viewed as overall moderate and vigorous PA, non-rural participants acquired significantly more total minutes of moderate PA (U = 8284.00, N1=139, N2 = 139, *p =* 0.038) and more total vigorous minutes of PA per week (U = 8320.50, N1=139, N2=139, *p =* 0.024). When PA was considered overall, non-rural participants were still significantly more likely to meet overall PA guidelines (*x*^2^(1) = 7.933, *p =* 0.005). 

### 3.2. Within-Group Results

We explored within-group differences between age and obesity categories for both rural and non-rural participants. Among rural participants, there was a statistically significant correlation between size or layout of indoor space negatively impacting PA during the COVID-19 pandemic and obesity with meeting PA guidelines (*x*^2^(1) = 6.304, *p* = 0.012). We found rural participants with obesity who reported size or layout of indoor space negatively impacting PA during the COVID-19 pandemic were 83.5% less likely to meet PA guidelines than rural participants with obesity who did not indicate size or layout of indoor space as negatively impacting PA (OR = 0.165, 95CI 0.037–0.729). 

Among non-rural participants with obesity, there were statistically significant correlations for wearing face masks negatively impacting PA (*x*^2^(1) = 4.975, *p* = 0.026) and concern for exposure negatively impacting PA (*x*^2^(1) = 4.600, *p* = 0.032) with meeting PA guidelines. Non-rural participants with obesity, who reported wearing face masks negatively impacting their PA, were 82% less likely to meet PA guidelines than those who did not indicate wearing face masks negatively impacting their PA (OR = 0.188, 95CI 0.041–0.860). Additionally, participants in this group concerned about exposure from PA were 84% less likely to meet PA guidelines than those who did not indicate concern about exposure to COVID-19 negatively impacting PA (OR = 0.167, 95CI 0.029–0.945). 

Statistically significant correlations for non-rural participants 35 years and younger were found between size or layout of indoor space negatively impacting PA and meeting PA guidelines (*x*^2^(1) = 6.032, *p* = 0.014), size or lack of yard negatively impacting PA and meeting PA guidelines (*x*^2^(1) = 6.262, *p* = 0.012), and concern about exposure to COVID-19 and meeting PA guidelines (*x*^2^(1) = 5.227, *p* = 0.022). Participants reporting size or layout of indoor space negatively impacting PA were 93% less likely to meet PA guidelines compared to those who did not indicate size or lack of indoor space as negatively impacting PA (OR = 0.077, 95CI 0.008–0.789). Non-rural participants who were 35 years and younger who indicated size or lack of yard negatively impacting PA were 92% less likely to meet PA guidelines than those who did not indicate size or lack of yard negatively impacting PA (OR = 0.088, 95CI 0.011–0.717). We found non-rural participants 35 years and younger who indicated concern about exposure to COVID-19 negatively impacting PA were 92% less likely to meet PA guidelines than participants who did not indicate concern for COVID-19 exposure as negatively impacting PA (OR = 0.087, 95CI 0.008–0.993) 

## 4. Discussion

Whether between groups or within group, the findings of the current study point to the COVID-19 pandemic having a greater impact on non-rural participants. There is a distinct feasibility that the difference between meeting PA guidelines were roughly the same during the pandemic as before, whereby rural participants were less likely to meet PA guidelines as compared to their non-rural counterparts. Pre-pandemic literature notes rural people being less likely to meet PA guidelines [24], in fact, meeting PA guidelines in rural areas lags behind urban areas by about 10 years [25]. It is important to note, when the data for this study were collected, the COVID-19 pandemic was just beginning to impact rural areas, while urban areas were already ravaged by the virus [30]. Therefore, non-rural areas likely had the most time to note impacts of the virus on their behavior. While there were no significant differences between groups in frequency, intensity, and time, it is important to note that, when viewed overall, 68% of participants did increase their frequency of PA, while the time and intensity both decreased. It is feasible that people were required to take more breaks during their day due to household responsibilities (e.g., tending to children/grandchildren, chores), which could have led to a higher frequency in PA. 

Having obesity in both rural and non-rural participants brought about barriers to PA [31]. Among rural people, a lack of indoor space being a barrier to meeting PA guidelines could be explained by the time in which the current survey was answered. The survey was administered during the height of the summer heat, which could have made participants think more about their indoor space. Conversely, obese non-rural participants viewed concern for exposure and mask wearing as a barrier to PA [32]. Perhaps non-rural people preferentially choose indoor fitness activities, which come with a greater risk. Obesity is a known risk factor for poor outcomes in COVID-19 [6,7,8,9]; therefore, going to indoor spaces to partake in PA would be of the utmost concern. At the time of data collection, gym and fitness facilities were allowed to loosen regulations nationwide. 

Counterintuitively, the youngest people in the non-rural group were the most concerned about exposure to COVID-19 while partaking in PA. A possible explanation would be younger people are the most likely to partake in exercise, thus exposing them to the virus. Cogently, indoor space and lack of a yard influenced the youngest of the non-rural group given urban and suburban areas would be least likely to have substantially large indoor and yard space sufficient to partake in common physical activities—a comparable finding to adolescents [33]. Furthermore, while the non-rural group made significantly more money than did the rural group, money may not go as far in urban areas due to cost-of-living differences; therefore, a lack of indoor and yard spaces in urban areas is logical. 

When considering PA at a more granular level, non-rural participants acquired more moderate PA, vigorous PA, and recreational PA. During the time of data collection, the cases in rural areas were beginning to climb while non-rural area cases were falling; simultaneously, gyms and other recreational facilities were beginning to open back up. Although non-rural area participants overall noted more barriers to PA, they were still more likely to partake in more PA at various intensities.

One of the most intriguing findings of the current study was found incidentally—all participants self-identified as rural, yet half were not according to their zip code (i.e., RUCC). The research team fielded many inquiries regarding people’s perception of themselves as rural, no matter where they may live during data collection. Participation in this study suggest a more fluid nature in the individual’s self-identity and self-expression as a rural person—using more socio-cultural definitions as opposed to physical size and location [34]. Defining rural here parallels race and gender whereby they are personally constructed, as opposed to labeled by outside entities [35]. This definition of rural may result in a widening of the gap for differences found between urban and rural individuals within this study. However, perhaps more importantly, these findings highlight the difficulty in defining rural through various governmental definitions and how the sociocultural implications of rurality may impact behaviors of individuals not identified as living in rural areas. There is wide variation and overgeneralization in the myriad definitions of rural [36]. Prior work of Gilbert et al. reported unique differences among rural communities in barriers to PA [37], indicating community size influenced available resources and infrastructure to support PA. Thus, differences in PA between urban and rural residents may have less to do with traditional definitions of rurality (i.e., population density, spatial location) and more with availability of community resources and the impact of social determinants of health, such as poverty.

### Limitations

The study design was cross sectional; therefore, there is a lack of follow-up which could loom large considering the ever-changing landscape of the virus. The sample was relatively small and homogenous; however, there was an even breakdown of participants between rural and non-rural places. We used subjective measures as a surrogate for meeting PA guidelines; in doing so, a large proportion of the sample (both rural and non-rural) met PA guidelines. However, it is commonplace to over-represent PA when using subjective measures [38,39]. While various components of the survey utilized were considered validated (e.g., GPAQ), many aspects of the survey had not been validated (e.g., frequency, access). While this was a limitation, it would have been impossible to completely validate a survey and put it through to the field in a timely manner. The survey was administered online, and it is impossible to know the true versus missing answers of the participants. For example, the participant was required to answer questions before moving on during the GPAQ; therefore, while the score still counts according to the GPAQ scoring, participants could not skip the question. If participants were allowed to skip the question, we would have had a better grasp on truly missing data. Finally, due to the relatively small sample size, effect sizes, and point-in-time data collection, the generalizability of these findings may be low. Conversely, this is the first study of its kind in an ever-evolving COVID-19 environment, which could be a basis for further inquiry. 

## 5. Conclusions

It may seem logical that non-rural areas would be impacted more from restrictions than would rural areas in relation to PA. In the current study, urban areas seemed to perceive more barriers inherent to the virus; however, rural areas were still less likely to meet PA guidelines during COVID-19. Non-rural area participants attained more moderate PA, vigorous PA, and recreational PA compared to rural participants. Lastly, it is important to understand “rural” as a socially identified construct as opposed to a black-and-white definition. People may forever identify as being “rural” even when they may not have lived in rural areas as defined by various entities for years. 

## Figures and Tables

**Table 1 ijerph-18-04991-t001:** Participant demographics split by rurality

Participant Characteristics	Rural N (%)	Rural N (%)	Total N (%)	*p*-Value
	139 (50.0)	139 (50.0)	365 (100.0)	
Age (Mean, SD) ^a^	46.99 ± 12.59	46.99 ± 12.59	45.54 ± 12.60)	0.061
BMI (Mean, SD) ^b^	30.73 ± 7.29	30.73 ± 7.29	29.68 (7.30)	0.002
Gender (Female) ^c^	125 (89.9)	125 (89.9)	254 (88.4)	0.368
Race (White) ^c^	135 (97.1)	135 (97.1)	265 (95.7)	0.255
Married ^c^	97 (69.8)	97 (69.8)	197 (71.4)	0.555
Income, ≥$50,000 ^c^	94 (69.1)	94 (69.1)	204 (75.8)	0.009

SD = standard deviation; BMI = body mass index. ^a^
*t*-test; ^b^ Mann-Whitney U; ^c^ chi-squared.

**Table 2 ijerph-18-04991-t002:** Physical activity variables split by rurality.

Physical Activity Variable	Rural *N* (%)	Non-Rural *N* (%)	Total *N* (%)	*p*-Value
Total Moderate Minutes PA/week ^a^ (Median, IQR)	90 (480)	180 (680)	142.50 (600)	0.038
Moderate Recreational Minutes PA/week ^a^ (Median, IQR)	0 (135)	60 (180)	35 (180)	0.054
Moderate Occupational Minutes PA/week ^a^ (Median, IQR)	0 (140)	0 (240)	0 (195)	0.462
Moderate Travel Minutes PA/week ^a^ (Median, IQR)	0 (0)	0 (0)	0 (0)	0.369
Total Vigorous Minutes PA/week (Median, IQR)	0 (90)	0 (210)	0 (180)	0.024
Vigorous Recreational Minutes PA/week ^a^ (Median, IQR)	0 (20)	0 (120)	0 (90)	0.006
Vigorous Occupational Minutes PA/week ^a^ (Median, IQR)	0 (0)	0 (0)	0 (0)	0.690
Meets PA Guidelines ^b^	72 (51.8)	95 (68.3)	167 (60.1)	0.005
Change in PA ^b^	87 (62.6)	95 (68.3)	182 (65.5)	0.377
Time ^b^				0.964
Increased	23 (28.75)	25 (29.07)	48 (28.92)	
Decreased	57 (71.25)	61 (70.93)	118 (71.08)	
Frequency ^b^				0.759
Increased	55 (69.62)	60 (67.42)	115 (68.45)	
Decreased	24 (34.29)	29 (32.58)	53 (31.55)	
Intensity ^b^				0.483
Increased	23 (29.87)	22 (25.0)	45 (27.27)	
Decreased	54 (70.13)	66 (75.0)	120 (72.72)	

IQR = interquartile range. ^a^ Mann-Whitney; ^b^ chi-squared.

## Data Availability

The data are not publicly available due to privacy concerns.

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
