# Peer review of "A Cross-Sectional Comparison of Physical Activity during COVID-19 in a Sample of Rural and Non-Rural Participants in the US"

_ijerph, 2021, doi:10.3390/ijerph18094991_

Round 1

Reviewer 1 Report

I would like to thank the Authors for improving the manuscript. I accept the manuscript  in the current form.

Author Response

Thank you for your suggestions

This manuscript is a resubmission of an earlier submission. The following is a list of the peer review reports and author responses from that submission.

Round 1

Reviewer 1 Report

The authors investigated physical activity patterns during the COVID-19 pandemic between people living in rural and non-rural areas. They concluded that the COVID-19 pandemic did not balance the gap of physical activity patterns between individuals living in rural and non-rural areas.

Abstract: I suggest to streamlining the Abstract; define the key aims of the study more precisely; describe in more details why answering to the research questions is important, and describe, if and how the present results expand upon the current literature.

“Participants self-identified as rural; however, when compared to their zip code, half were determined to have not been rural by their Rural Urban Continuum Code.”; is this sentence/information important to the understanding of the ‘plot’ of the Abstract?

Likewise: “Conversely, non-rural participants were more likely to have a lower BMI, higher income, higher education, and reported more barriers to PA.” is this sentence not simply the opposite of what the preceding sentence was telling?

“There was no difference in the perception of COVID-19’s impact on PA, specifically. however, rural participants were significantly less likely to meet cardiorespiratory PA recommendations compared to non-rural participants. )”; delete the parenthesis; was meeting cardiorespiratory physical activity recommendations an aim of the study?

“This study demonstrates the continued disparity in PA between rural and non-rural residents, despite the supposition of COVID-19 being less impactful in rural areas due to sparse populations.”; ok, but was this the aim of the present study? And: was COVID-19 less impactful because in rural areas the population is sparse, or perhaps because the physical activity patterns of individuals in rural areas showed floor effects?

“Feasibly, people grew up in rural areas, but now live in metropolitan areas, may continue to identify as rural.” How should this claim explain the pattern of results?

Introduction:

First paragraph: “In the early  ….  penalty”; while this information might be interesting, it appears disconnected from the title and the Abstract. Likewise, it was difficult to me to figure out what might be the implicit or explicit information of this paragraph.

Wording: “leading the way”; “..There is hope from approved vaccines; however, it will still take some time to get them to the masses”; while English is not my mother tongue, my impression is that the style appears too colloquial and informal; such a style might be appropriate for a tabloid, but not for a scientific publication. Further, I had difficulties to understand, why the sentence regarding vaccines is important. Perhaps the sentence is misplaced.   

“Rural communities tend to be older, obese, have higher rates of smoking, and have more comorbidities compared to urban areas”; I do not think that urban areas could be compared to people, communities or individuals. Perhaps the authors wanted to say that compared to individuals living in urban areas, individuals living in rural areas are older, show a higher BMI, report a higher prevalence of (tobacco?) smoking and report more somatic (?) and psychiatric (?) comorbidities.

“These very same characteristics are also associated with poor outcomes in COVID-19”; but which ‘very same characteristics’ are the authors referring to? Likewise, define in more details what ‘poor outcomes in COVID-19’ might be; the authors should be much more specific and precise.

Overall, the Introduction section needs a thorough revision; topics are presented in a scattered fashion; information and statements appear blurred; the theoretical concepts of the study must be described in a thorough fashion; it remains unclear what the scientific community already knows about the topics (what is the state-of-the-art) and how and to which extend the present data expand upon previous studies.

Material and methods: please describe the recruitment procedure in more details; “(e.g., Facebook)” is not sufficient. Does the use of social network sites (SNSs) not implicitly assume the ability to read? How did the authors operationalize the inclusion criterion “to be physically active”? being physically active may range from gardening, strolling around with the dog, bringing children to kindergarten/school to striving for a good results at a marathon competition.

Demographics; from a statistical point of view, it is highly discouraged to categorizing continuous variables into ordinal/nominal variables (MacCallum et al., 2002); please consider this issue when running the statistics.

Statistics: it should be: SPSS® 25.0 (IBM Corporation, Armonk NY, USA) for Windows®/Apple Mac®. “….differences in meeting PA guidelines both….”; was ‘meeting PA guidelines” an aim of the study?

I assume that the level of significance was set at alpha < .05, though, this should be explicitly mentioned. Further, I strongly recommend to run a series of correlations to calculate the associations between age, BMI, and physical activity indices, to name a few. Please also report effect sizes.

Results: Table 1; do not use vertical bars. Reporting just a p-value is not meaningful without the sample size, degrees of freedom, and t-values (Wasserstein et al., 2019).

Table 1; I suggest to structuring the items in a more appropriate fashion; first, all metric variables (with M (SD) and t-values and p-values), then, X2-results (with N, df, X2-values and p-values). The authors should help the reader to get a quick and precise overview.

Table 1: Data suggest that individuals in rural areas are descriptively older, compared to individuals in non-rural areas; however, the effect size (Cohen’s d) was d = 0.11 (trivial effect size); likewise, data suggest that individuals in rural areas have a higher BMI, compared to individuals in non-rural areas; however, the effect size (Cohen’s d) was d = 0.14 (trivial effect size); as such, there are no age and BMI differences between the two groups.

Overall, the pattern of results of Table 1 suggests that group differences are spurious and trivial; as such this result does not match what the authors claim in the Abstract section.

Discussion; based on the statistical issues mentioned above, it appears that the Discussion section needs a thorough revision.

References

MacCallum, R.C., Zhang, S., Preacher, K.J., Rucker, D.D., 2002. On the practice of dichotomization of quantitative variables. Psychol Methods 7(1), 19-40.

Wasserstein, R.L., Schirm, A.L., Lazar, N.A., 2019. Moving to a World Beyond “p < 0.05”. The American Statistician 73(sup1), 1-19.

Reviewer 2 Report

I would like to thank the Authors to prepare a manuscript about physical activity during the pandemy of Covid-19. The topic is very actual. In connection with my responsibilities, there are many issuess to discuss. My opinion is presented below.

The authors write in the abstract that: "The purpose of this study was 9 to determine COVID-19's impact on PA in rural residents", but the title of the work points to a different study. The authors should improve the title of the work, that would be consistent with the main aim of the work. A group of people from non-rural areas will be the control group in this study.

In the abstract, description of research method is insufficient, there is only information about online survey - there is no detailed information whether the survey was based on some standardized questionnaire, which would be advisable.
The authors present only data obtained during the pandemic, is it possible to compare with the pre-covid period?

Introduction
Please remove large paragraphs, improve text editing in connection with journal guidelines, please justify text, standardize paragraphs.
The authors describe rural areas and urban areas only of the United States (US) - which should also be included in the title of the work.
In the introduction, the authors write abou mortality rates, which is not related to the aim of this study. The introduction should be strictly related to the topic under consideration. The authors should explain what are the differences between PA in the rural and urban areas. What are their assumptions about changing this situation during Covid pandemy.

The aim of the work from the abstract differs from aim which is included at the end of the introduction.
Setnece "COVID-19 is not going away anytime soon ...." It is not recommended to be in the aim of the study section.
The inclusion criteria: age> 18 years of old is to wide. The authors did not focus on the study of a specific age population group, e.g. young adults or eldery people. Comparing a student with f.ex. a retired person distorts the overall results and conclusions of the study. Analyzes should be performed separately on these groups.
In the parapraph "Setting"- there is no information about how many people participated in the study, how many successfully completed it. 
The authors asked only about the pandemic period, they do not have data that relates to pre-pandemic physical activity.
In the statistical analysis, they do not describe what value was considered as a statistically significant.
Legends should appear below the tables. 
Graphic text editing is needed.
References (see Table 1) .- not preferred in scientific articles, please remove the word "see".
The presented data in the Table 1 is very poor. There are no detailed analyzes. The presented results can't be published in the current form. Manuscript requires a major revisions.

Discussion
The discussion is too short, it should be more detailed and refer to the latest reports on this topic. 
Conclusions- do not correspond to the aims of the study. They show the opinions of the authors rather than the conclusions of the obtained results.
Literature:
Authors should add more current referneces published mainly in 2020 and 21 -in relation to the physical activity during the pandemic period.